# Symmetry of the Vermillion Height after Modified Rotation-Advancement Cheiloplasty

**DOI:** 10.3390/jcm11226744

**Published:** 2022-11-15

**Authors:** Ivy Valdez Tangco, Kishor Bhandari, Chuan-Fong Yao, Ting-Chen Lu, Philip Kuo-Ting Chen

**Affiliations:** 1Department of Otolaryngology, Head and Neck Surgery, Baguio General Hospital and Medical Center, Baguio 2600, Philippines; 2National Academy of Medical Sciences, Bir Hospital, Kathmandu 44600, Nepal; 3Craniofacial Center, Department of Plastic and Reconstructive Surgery, Chang Gung Memorial Hospital, Chang Gung University, Taoyuan 333, Taiwan; 4Craniofacial Center, Department of Plastic and Reconstructive Surgery, Taipei Medical University Hospital, Taipei 110, Taiwan

**Keywords:** modified rotation advancement cheiloplasty, vermillion height, vermillion length, unilateral cleft lip repair

## Abstract

(1) Background: This study aimed to determine the postoperative vermillion symmetry between the cleft and non-cleft sides of patients with unilateral cleft lip during the early and late postoperative periods. (2) Methods: 57 patients with complete and 38 with incomplete unilateral cleft lips operated on between 2010 and 2014 were retrospectively evaluated within 1 month (T1), 9 months to 1 ½ years (T2), and more than 4 years (T3). Vermilion heights of the cleft and non-cleft sides were measured from frontal photographs. The Cleft Lip Component Symmetry Index (CLCSI) was used to determine the symmetry of the cleft and non-cleft sides and was then analyzed. (3) Results: Among the 95 patients studied, vermilion height was excessive on the cleft side throughout the three time periods. There was a significant increase in CLCSI from T1 to T2 for both complete and incomplete types, and a significant increase from T1 to T3 only in the incomplete group and no difference from T2 to T3 for both the groups. (4) Conclusions: Even with efforts to obtain a symmetric vermilion height during the primary cheiloplasty, vermilion height excess was noted with time in complete and incomplete cleft types. Secondary revisional vermilion surgery may be performed to achieve symmetry.

## 1. Introduction

The appearance after cleft lip repair is believed to significantly affect the patient’s social competence and personality development in the long term [1]. Due to these effects, cleft lip repair has become highly specialized in recent years, and aggressive efforts have been put to improve the aesthetic outcomes.

The appearance of the nasolabial region is an important measure of surgical treatment success and thus influences the degree of facial attractiveness and psychosocial relations. Residual deformities of cleft-lip repair include minor irregularities and vermilion asymmetry, which are stigmata of postsurgical cleft repair [2]. Vermilion asymmetry is quite often noticeable in the postoperative period and is a frequent reason for secondary deformity [3]. The vermillion color makes it more obvious than the upper lip scar, which is usually white and flat. Very few studies have focused on the symmetry of the vermillion height. Upon observation, the vermilion height over the cleft side appeared to be excessive in the long term, especially in incomplete cleft lip cases. In addition, the vertical discrepancy of the upper lip measurement appears more obvious to the naked eye and is readily noted as a facial disfigurement rather than a lateral lip length discrepancy [4].

Previous studies have investigated the growth of the lip after cheiloplasty through anthropometric measurements using the extended Mohler, Tennison–Randall, Millard, and LeMesurier techniques [5,6,7]. In our center, the modified rotation advancement [8] for unilateral cleft lip repair incorporates Mohler’s incision, the small, lower triangular flap, and the vermilion triangular flap [9]. Outcome studies of the modified rotation advancement cheiloplasty show the vermillion height is higher in both complete and incomplete lips at the 1-year follow-up [8]. However, a longer follow-up result was not investigated. This study was undertaken to evaluate the vermilion height symmetry of patients with unilateral cleft lip who underwent primary cheiloplasty using the modified rotation advancement with longer follow-ups.

Cleft and normal side vermilion height of the same patients during the three time periods was measured within the first month after surgery, at 9–12 months, and at >4 years postoperatively. The change in the vermilion thickness was also compared between the complete and incomplete cleft lip groups after repair.

## 2. Materials and Methods

The current study was approved by the Institutional Review Board of Chang Gung Medical Foundation (IRB number 202100550B0). Patients with unilateral complete and incomplete cleft lip between 2010 and 2014 were included as per the following criteria: (1) 3-month-old babies with complete or incomplete unilateral cleft lip who underwent repair utilizing the modified rotation advancement cheiloplasty; and (2) patients with complete postoperative frontal view photos during the immediate postoperative (1 week to 1 month), one year, and more than 4 year periods. All unilateral cheiloplasties were performed by the senior author (P.K.-T.C.).

Exclusion criteria were as follows: (1) patients with incomplete follow-up and photographs; (2) patients who had a minor form, microform, or mini-microform unilateral cleft lip; (3) patients with any form of cleft lip on the other side; (4) patients with other craniofacial abnormalities; (5) patients with postoperative complications, such as infection, bleeding, or wound dehiscence; and (6) patients with subsequent operations on the upper lip vermilion following primary cheiloplasty.

### 2.1. Surgical Techniques and Study Methods

The modified rotation advancement cheiloplasty incorporated Mohler’s incision, triangular flap, and triangular vermilion flap for unilateral cleft lip repair. Landmarks such as the crista philtri inferior right/left (CPHIR/CPHIL), labrale superius (LS), and red lines were noted. The Noordhoff’s point, which is a distinct point where the white skin roll changes direction and is also where the vermillion first widens is identified, was taken as a reference in creating the vermillion flap. Excess mucosa or muscle on the free border of the vermilion was trimmed so that the two edges fit together perfectly [10]. 

Preoperative preparation was performed for all patients as indicated, such as presurgical orthodontic evaluation, nasoalveolar molding, and preoperative lip taping. Lip taping and silicon sheet application on the wound area after suture removal were performed until 6 months postoperatively.

Digitized photographs of the patients who were followed up immediately (1 week to 1 month) (T1), at 9 months–1½ years (T2), and more than 4 years (T3) after primary cheiloplasty, were obtained for reference. The pictures were taken with the patients in frontal view using a camera (Canon EOS600D, 18 megapixels) in a 1:1 ratio. An attempt was made for the photos to be taken in a natural head position as much as possible. The same institutional professional photographer took the photos. The written consent was taken from the parents for the study. The outcomes of cheiloplasty repair were all measured using Adobe Photoshop Version 13.1.2 (Adobe System, San Jose, CA, USA) by two observers (two trained research assistants). Only pictures in which the lips relaxed without facial expression such as crying or laughing were included.

A total of 191 patients underwent the modified rotation advancement cheiloplasty from 2010 to 2014. Of these, 96 patients were excluded, leaving 95 eligible for the assessment.

#### 2.1.1. Landmarks and Measurement of Vermilion Height Symmetry

The pictures were initially adjusted to achieve symmetry by creating a horizontal line through both the medial canthi or both the lateral canthi. The standard landmarks were then identified based on research by Farkas, such as the CPHIR (junction between the upper lip vermilion and right philtral peak) and the CPHIL (junction between the upper lip vermilion and left philtral peak) [11].

#### 2.1.2. Measurement of Upper Lip Vermilion Height

The vermilion heights defined by previous authors [5,6,7] were adopted in the present study. The vermilion height cleft side (VHC) was measured from the CPHIL to the vermilion free edge, and the vermilion height non-cleft side (VHNC) was measured from the CPHIR to the vermilion free edge, as shown in Figure 1.

Three time intervals (T1, T2, and T3) were considered for comparison of the same patient for both the incomplete and complete cleft types (Figure 2 and Figure 3).

### 2.2. Index of Comparison of the Cleft and Non-Cleft Sides

All upper lip anthropometric outcomes were assessed using the Cleft Lip Component Symmetry Index (CLCSI) [7]; a dimensionless quotient (Q) for the corresponding areas or distances was measured; the cleft side (C) was divided by the non-cleft side (NC) and multiplied by 100; thus, Q = C/NC × 100. The values were compared with one another independent of photo magnification. A symmetry quotient of 100 indicates perfect symmetry. A value >100 meant that the cleft side was larger or longer than the non-cleft side, and that <100 meant that the cleft side was smaller or shorter than the non-cleft side7. The vermilion height was considered asymmetric if the percentage fell >5% of 100, meaning that >105% was excessive and <95% was deficient.

#### Statistical Analysis

Anthropometric measurements were analyzed using the Statistical Package for Social Sciences (SPSS) Version 17 (SPSS Inc., 2008, IBM Corp., Armonk, NY, USA). Interobserver reliability was initially determined using the intraclass correlation coefficient (ICC). The means for the upper lip VHC and VHNC ratios were initially determined. The measurements for the incomplete and complete cleft types were compared using paired *t*-test for intragroup comparison. Levene’s test for equality of variance was nonsignificant; hence, intergroup comparison between the incomplete and complete cleft lip groups was performed using an independent t-test. The statistical significance level was set at a confidence interval of 95% (<0.05).

## 3. Results

Of the 95 patients included in the study, 57 had complete cleft lip and 38 had incomplete cleft lip. Fifty-nine males and 36 females were included, with 73 left-sided and 22 right-sided cleft lip, respectively. The mean follow-up period for all patients was as follows: T1, 17.39 days; T2, 12.22 months; T3, 6.16 years (Table 1).

Interobserver reliability using ICC was 0.941, hence interpreted as excellent [12].

The CLCSI mean values for T1, T2, and T3 show excess vermilion height for the incomplete and complete groups. Intragroup comparisons for VH (vermillion height) for T1, T2, and T3 were compared using a paired *t*-test (Table 2). In the incomplete cleft lip group, there was a statistically significant increase from T1 to T2 (*p* = 0.001) of 5.675% and from T1 to T3 (*p* = 0.027) of 3.826%, but not between T2 and T3 (*p* = 0.325). As for the complete cleft group, there was a significant increase from T1 and T2 (*p* = 0.001) of 6.516%, but there was no significant difference between T2 and T3 (*p* = 0.105) and T1 and T3 (*p* = 0.069).

The results of the independent *t*-test comparing the incomplete and complete cleft lip groups revealed no significant difference in the vermilion height means during T1, T2, and T3 (Table 3).

## 4. Discussion

Facial symmetry is one of the goals of cleft lip surgery, and objective evaluation of postoperative lip outcomes is of utmost importance. Anthropometric measurements of vermilion length, lip height, and philtrum height are key to evaluating the facial symmetry of cleft lip repair [7]. The study focused on the postoperative vermilion height measurement because this is a common observation after unilateral cleft lip repair, and parents have noticed discrepancies in vermilion height rather than other anatomic measures.

The modified rotation advancement for unilateral cheiloplasty uses a definite anatomic point, which is the base of the cleft side philtral column or the cupid’s bow peak (CPHI). Identifying the cupid’s bow peak on the cleft side is somewhat controversial, although one of the most accepted and surgically practiced concepts at present includes the technique described by Noordhoff [9], in which there is adequate vermilion height, usually matching the VHNC cupid’s bow and a good-quality cutaneous roll. This is an important anatomic point that can only be moved when there is a severe discrepancy in horizontal measurements between the cleft and non-cleft sides of the lips. Moreover, the technique uses a vermillion triangular flap at the junction between the wet and dry mucosa, which is meticulously tailored to address the vermilion mismatch that would otherwise result from other techniques [13,14]. 

The goal of cheiloplasty is to achieve “normal” symmetry, which is asymmetric in actuality [5]. In this sample population, the vermilion height categorization being deficient, or excess was adopted [5], considering that in normal individuals some degree of lip asymmetry is expected. The patients’ vermillion height in the study was in excess throughout the three time periods, with a significant increase from T1 to T2 of 5.675% and 6.516% for the incomplete and complete groups, respectively. In addition, between T1 and T3, the vermilion height in the incomplete group significantly increased to 3.826%, whereas no significant increase was found in the complete group.

Preoperatively, the lateral lip of cleft patients in the present study was noted to bulge compared with that on the non-cleft side, especially among patients with incomplete cleft lip. This vermilion deformity might have been due to mucosal problems, muscle problems, or a mixture of components. A common mistake might be to include too much muscle in the vermilion flap, hence the bulging appearance in the free border. Keeping the same volume at the base of the vermilion as its counterpart below the CPHR point is important to avoid this problem [10]. No significant differences in the vermilion height were observed, suggesting adequate vermilion trimming in the CLCSI for the vermillion height of T1, T2, and T3.

Regarding the excess vermilion height measurements noted in our study, other studies obtained similar postoperative findings of excess vermilion. A study by Kluba et al. found that an excess of approximately 17% may be found in patients who have undergone Tennison–Randall repair at 4 years of age [1]. This vermilion excess may be attributed to either a systematic problem of the applied surgical technique or anatomic irregularities (i.e., increased number of submucosal salivary glands in the lateral lip element on the cleft side). In contrast, a study by Amaratunga et al., which compared the Millard and LeMesurier techniques, observed that the former technique produced a shorter vermilion height 1 month after surgery over time, although it significantly improved 1 year after surgery. Millard has accepted that the method produced an asymmetric vermilion such that he designed the “V-Y plasty” as a secondary procedure [7].

In our study, patients followed up for 9 months to 1.5 years (T2) had the most excess vermilion, and a significant difference was found in the immediate postoperative period (T1) in both complete and incomplete cleft patients. This may be due to scar tissue overgrowth over the vermilion or prolonged swelling or tissue reaction. After at least 4 years of follow-up, this condition gradually subsided, and less vermilion excess was noted in T3 (longer than T3). In T3, the complete cleft lip cases had a less vermilion height compared to that in the incomplete cleft lip cases. The incomplete cleft lip cases tended to have excess vermilions even after 4 years of follow-up and had more tissue over the vermilion, which may explain this condition. 

In our observation, the vermilion height changed dynamically. It may become excessive postoperatively and possibly less after longer follow-up; however, during the follow-up period in our study, no patient had a reduced vermilion height during the follow-up period and immediately after surgery. However, the patients were not followed up for more than 10 years, and we cannot formulate a conclusion after 10 years of follow-up.

Surgeries to correct secondary deformities should be planned in a timely manner, especially in a growing child. Parents should be informed that these deformities can be improved, bearing in mind that the successful correction of secondary deformities is frequently more challenging than the primary repair. An expected pattern of growth and development of each facial component must be considered, and any interference with growth must be minimized [1]. With the outcomes noted in this study, parents may be counselled that improving the vermilion height might not be significant from the operative time up to 4 years of age, especially in patients with incomplete cleft. Secondary vermilion correction might be performed with vermilion trimming over the cleft side or augmentation with a dermofascia graft over the normal side to balance the vermilion height depending on the overall vermilion appearance, which can be performed before elementary school or during alveolar bone grafting. However, so far, there was no patient asking for revision of the vermillion. 

The limitation of this study is that patients were not classified according to whether they had a preoperative alveolar cleft. Preoperative severity of the cleft gap was not noted and no longer included in the scope of this study. The results of the technique during operation would have been a better basis for the subsequent measurements and therefore would have provided a better understanding of the dynamic changes of the vermilion over time.

## 5. Conclusions

This study demonstrated vermilion height excess in both the complete and incomplete cleft lip types operated utilizing the modified rotation advancement cheiloplasty from the early follow-up period up to 7 years of age. Improvements after this time period up to 4 years of age can be found in complete and incomplete cleft patients, but the improvement is not significant. Long-term follow-up is needed to further understand the change in vermilion. Secondary correction of the vermilion height can be performed during alveolar bone graft repair or at any time if the patient is concerned with minimizing future psychosocial problems.

## Figures and Tables

**Figure 1 jcm-11-06744-f001:**
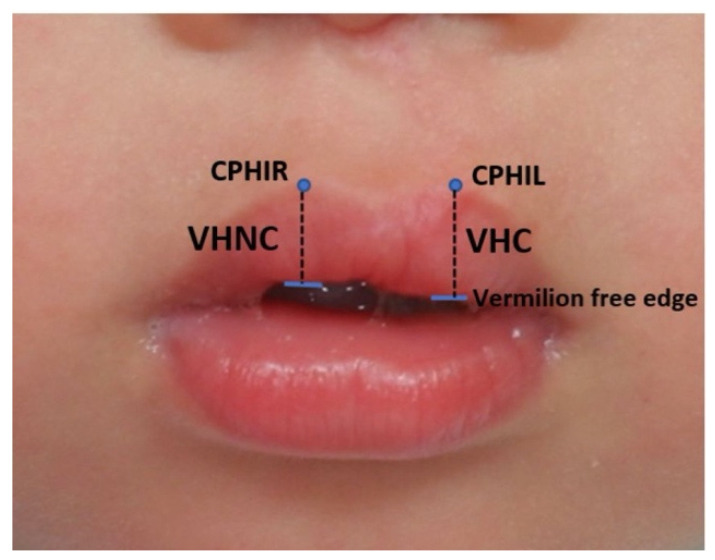
Landmarks and measurement of the upper lip vermilion height. A one-year-old boy who underwent cheiloplasty for an incomplete cleft lip. Landmarks: crista philtri inferior right (CPHIR), crista philtri inferior left (CPHIL), and vermilion-free edge (indicated by the horizontal blue line). Measurements: VHC, vermilion height cleft side; VHNC, vermilion height non-cleft side.

**Figure 2 jcm-11-06744-f002:**
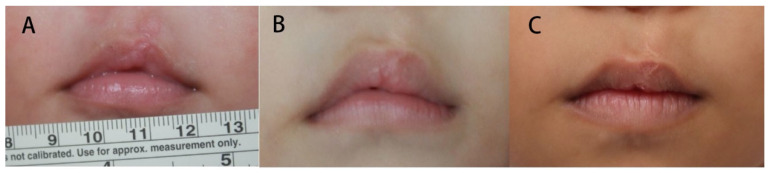
Vermillion change of an incomplete cleft lip. The vermillion was deficient compared with the normal side 1 month after surgery; however, it became excessive at 11 months and gradually subsided at 7 years old. (**A**) postoperative 1 month. (**B**) 11 months old. (**C**) 7 years old.

**Figure 3 jcm-11-06744-f003:**
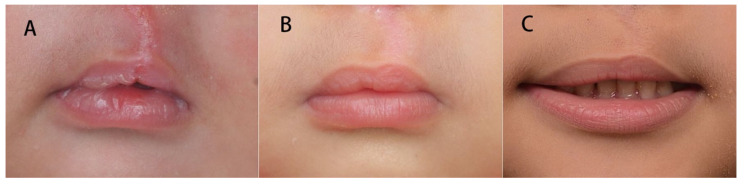
Vermillion change of a complete cleft lip patient. The vermillion was equal to that of the normal side 1 week postoperatively. At 12 months of age, the vermillion became excessive compared with the normal side. It gradually subsided at 5 years of age. (**A**) postoperative 1 week. (**B**) 12 months old. (**C**) 5 years old.

**Table 1 jcm-11-06744-t001:** Demographic data and follow-up.

	Complete Cleft Lip	Incomplete Cleft Lip	*p*-Value
Patient Number	57	38	>0.05
Male/female	31/26	10/27	
Cleft side (Left/Right)	26/46	10/27	
Mean follow-up			>0.05
T1 (days)	17.74 (7–30)	16.87 (7–30)	
T2 (months)	11.98 (9–15)	12.58 (9–15)	
T3 (years)	6.42 (5–9)	5.78 (5–9)	
Cleft Lip Component Symmetry Index (CLCSI)	Mean (%)	Mean (%)	
T1	108.69	106.25	0.36
T2	115.20	111.93	0.189
T3	111.94	110.08	0.486

**Table 2 jcm-11-06744-t002:** Changes in vermillion height at T1, T2, and T3.

Cleft Type	Time Interval	VH ^†^ Mean (%)	ΔVH Mean (%)	Standard Deviation	Standard Error Mean	*p* Value
Incomplete	T1 T2	106.25 111.93	+5.675	9.966	1.617	0.001 *
	T2 T3	111.93 110.08	−1.848	11.420	1.853	0.325
	T1 T3	106.25 110.08	+3.826	10.227	1.659	0.027 *
Complete	T1 T2	108.69 115.20	+6.516	14.332	1.898	0.001 *
	T2 T3	115.20 111.94	−3.2625	14.936	1.978	0.105
	T1 T3	108.69 111.94	+3.254	13.236	1.753	0.069

^†^ VH: vermillion height, Δ: difference in vermillion height, * significant.

**Table 3 jcm-11-06744-t003:** Comparison of vermillion height means at T1, T2, and T3 between incomplete and complete cleft types.

Time Period	Cleft Type	VH ^†^ Mean (%)	ΔVH Mean (%)	Standard Error Difference	*p* Value
T1	Incomplete	106.25	+2.433	2.644	0.360
	Complete	108.69			
T2	Incomplete	111.93	+3.275	2.582	0.189
	Complete	115.20			
T3	Incomplete	110.08	+1.860	2.662	0.486
	Complete	111.94			

^†^ VH: vermillion height, Δ: difference in vermillion height.

## Data Availability

The relevant data will be readily made available by the corresponding author when asked upon.

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
