# Peer review of "Symmetry of the Vermillion Height after Modified Rotation-Advancement Cheiloplasty"

_jcm, 2022, doi:10.3390/jcm11226744_

Round 1

Reviewer 1 Report

Basic reporting

Out of 95 participants, there were 59 men and 36 women. The study also includes children and babies. Hence it would be better to mention the gender as male or female. Lines 133 to 134

The study is well written and makes for an interesting read. The language used is precise, clear and hardly any grammatical errors.

Experimental design

Please provide information regarding consent, if taken from parents. Please provide information regarding positioning of the head when photographs were taken ?Was there a professional photographer ? Who took the photos ?

Was the same camera used in all patients throughout the entire time frame? The entire study would take at least eight years. Were the photos taken by the same person?

This particular study does not mention the age of the participants. Were there any age related differences with the results of the surgery? What was the age of the oldest patient?

How many patients required secondary repairs after the follow up ?

Validity of findings

The study presents important objective results whereby the healthcare providers can provide information regarding the changes expected after surgery and clinical decisions regarding secondary repair. Do the authors recommend that the surgeons postpone any secondary repair until 4 years of age? What are the implications for practicing clinicians?

Author Response

Reviewer 1:

Point 1: it would be better to mention the gender as male or female. Lines 133 to 134

Response 1: Thank you for your recommendation. The terms were changed.

Point 2: Please provide information regarding consent, if taken from parents. Please provide information regarding positioning of the head when photographs were taken ? Was there a professional photographer ? Who took the photos?

Was the same camera used in all patients throughout the entire time frame? The entire study would take at least eight years. Were the photos taken by the same person?

Response 2: Thank you for your thoughtful consideration. The written consent was taken from the parents for the study. The photos were attempted to be taken in natural head position as much as possible. The same institutional professional photographer took the photos. The following sentences were added in line 92-94

“The photos were attempted to be taken in natural head position as much as possible. The same institutional professional photographer took the photos. The written consent was taken from the parents for the study.”

Point 3: This particular study does not mention the age of the participants. Were there any age related differences with the results of the surgery? What was the age of the oldest patient?

Response 3: Thank you very much for the suggestions. All the cleft lip surgeries performed in our hospital was at the age of 3-month-old. (The information has been changed in line 66). The photos taken for the patients were quite regularly in our institute. We compared the measurements in different time frame in a single patient. The different time frame is T1 (range from 7 to 30 days), T2 (range from 9-15 months) and T4 ( range from 5-9 years). The range were added in Table 1. In our observation, the change of the vermillion is very slow. Therefore, the range of the age should not make an obvious difference.

Point 4: How many patients required secondary repairs after the follow up ?

Response 4: Thank you for your questions. In this paper, we collect the patients underwent rotation advancement cheiloplasty between 2010-2014. The patient is around 8- 12 years old right now. So far, no patient asks about secondary revision of the vermillion. This information was added to line 250.

Point 5: Do the authors recommend that the surgeons postpone any secondary repair until 4 years of age? What are the implications for practicing clinicians?

Response 5:  Thank you for your thoughtful question. In this study, it seems that the excess of the vermillion disappears very slow. However, the vermillion excess gradually resolving after 4 years is promising. We do suggest follow up and perform secondary revision upon request of the patients.

Reviewer 2 Report

This study is describing the postoperative vermilion symmetry between the cleft and non-cleft sides of patients with unilateral cleft lip, and its change over a long period of time.

I think it is a very important paper that presents in an objective way the phenomenon that many experienced cleft surgeons have felt first-hand. Although I do not think that the cause of the increase of vermilion height is clarified by the discussion of the present study, that is a different issue from the essence of this study.

I would like to make only one minor point. In P2 L95, the author mentioned “Only pictures in which the teeth were occluded and~”. How did you determine the eligibility of the photographs in the case of patients with unerupted teeth (T1)? I think that ensuring the eligibility of the photos is very important in judging the value of this paper, so I think it would be better to add more details.

As for the rest, I found it a genuinely interesting study.

Author Response

Reviewer 2

Point1: In P2 L95, the author mentioned “Only pictures in which the teeth were occluded and~”. How did you determine the eligibility of the photographs in the case of patients with unerupted teeth (T1)? I think that ensuring the eligibility of the photos is very important in judging the value of this paper, so I think it would be better to add more details.

Response 1:

Thank you very much for your thoughtful suggestion. We will change the sentences in the paper,

"Only pictures in which the teeth were occluded and lips relaxed were included." will be changed to " Only pictures in which the lips relaxed without facial expressions such as crying or laughing were included" 
